# LABEL EMBEDDING NETWORK:
# LEARNING LABEL REPRESENTATION FOR SOFT TRAINING OF DEEP NETWORKS

## ABSTRACT

We propose a method, called Label Embedding Network, which can learn label representation (label embedding) during the training process of deep networks. With the proposed method, the label embedding is adaptively and automatically learned through back propagation. The original one-hot represented loss function is converted into a new loss function with soft distributions, such that the originally unrelated labels have continuous interactions with each other during the training process. As a result, the trained model can achieve substantially higher accuracy and with faster convergence speed. Experimental results based on competitive tasks demonstrate the effectiveness of the proposed method, and the learned label embedding is reasonable and interpretable. The proposed method achieves comparable or even better results than the state-of-the-art systems.

## 1 INTRODUCTION

Most of the existing methods of neural networks use one-hot vector representations for labels. The one-hot vector has two main restrictions. The first restriction is the "discrete distribution", where each label is distributed at a completely different dimension from the others. The second restriction is the "extreme value" based representation, where the value at each dimension is either 1 or 0, and there is no "soft value" allowed. Those deficiencies may cause the following two potential problems.

First, it is not easy to measure the correlation among the labels due to the "discrete distribution". Not being able to measure the label correlation is potentially harmful to the learned models, e.g., causing the data sparseness problem. Given an image recognition task, the image of the *shark* is often similar to the image of the *dolphin*. Naturally, we expect the two labels to be "similar". Suppose that we have a lot of training examples for *shark*, and very few training examples for *dolphin*. If the label *shark* and the label *dolphin* have similar representations, the prediction for the label *dolphin* will suffer less from the data sparsity problem.

Second, the 0/1 value encoding is easy to cause the overfitting problem. Suppose $A$ and $B$ are labels of two similar types of fishes. One-hot label representation prefers the ultimate separation of those two labels. For example, if currently the system output probability for $A$ is 0.8 and the probability for $B$ is 0.2, it is good enough to make a correct prediction of $A$. However, with the one-hot label representation, it suggests that further modification to the parameters is still required, until the probability of $A$ becomes 1 and the probability of $B$ becomes 0. Because the fish $A$ and the fish $B$ are very similar in appearance, it is probably more reasonable to have the probability 0.8 for $A$ and 0.2 for $B$, rather than completely 1 for $A$ and 0 for $B$, which could lead to the overfitting problem.

We aim to address those problems. We propose a method that can automatically learn label representation for deep neural networks. As the training proceeds, the label embedding is iteratively learned and optimized based on the proposed label embedding network through back propagation. The original one-hot represented loss function is softly converted to a new loss function with soft distributions, such that those originally unrelated labels have continuous interactions with each other during the training process. As a result, the trained model can achieve substantially higher accuracy, faster convergence speed, and more stable performance. The related prior studies include the traditional label representation methods (Hardoon et al., 2004; Hsu et al., 2009; Bengio et al., 2010), the "soft label" methods (Nguyen et al., 2014), and the model distillation methods (Hinton et al., 2014).

Our method is substantially different from those existing work, and the detailed comparisons are summarized in Appendix E. The contributions of this work are as follows:

- **Learning label embedding and compressed embedding**: We propose the Label Embedding Network that can learn label representation for soft training of deep networks. Furthermore, some large-scale tasks have a massive number of labels, and a naive version of label embedding network will suffer from intractable memory cost problem. We propose a solution to automatically learn compressed label embedding, such that the memory cost is substantially reduced.

- **Interpretable and reusable**: The learned label embeddings are reasonable and interpretable, such that we can find meaningful similarities among the labels. The proposed method can learn interpretable label embeddings on both image processing tasks and natural language processing tasks. In addition, the learned label embeddings can be directly adapted for training a new model with improved accuracy and convergence speed.

- **General-purpose solution and competitive results**: The proposed method can be widely applied to various models, including CNN, ResNet, and Seq-to-Seq models. We conducted experiments on computer vision tasks including CIFAR-100, CIFAR-10, and MNIST, and on natural language processing tasks including LCSTS text summarization task and IWSLT2015 machine translation task. Results suggest that the proposed method achieves significantly better accuracy than the existing methods (CNN, ResNet, and Seq-to-Seq). We achieve results comparable or even better than the state-of-the-art systems on those tasks.

## 2 PROPOSED METHOD

A neural network typically consists of several hidden layers and an output layer. The hidden layers map the input to the hidden representations. Let's denote the part of the neural network that produces the last hidden representation as

$$\boldsymbol{h} = f(\boldsymbol{x}) \tag{1}$$

where $\boldsymbol{x}$ is the input of the neural network, $\boldsymbol{h}$ is the hidden representation, and $f$ defines the mapping from the input to the hidden representation, including but not limited to CNN, ResNet, Seq-to-Seq, and so on. The output layer maps the hidden representation to the output, from which the predicted category is directly given by an $\mathrm{argmax}$ operation. The output layer typically consists of a linear transformation that maps the hidden representation $\boldsymbol{h}$ to the output $\boldsymbol{z}$:

$$\boldsymbol{z} = o(\boldsymbol{h}) \tag{2}$$

where $o$ represents the linear transformation. It is followed by a softmax operation that normalizes the output as $\boldsymbol{z}'$, so that the sum of the elements in $\boldsymbol{z}'$ is 1, which is then interpreted as a probability distribution of the labels:

$$\boldsymbol{z}' = \mathrm{softmax}(\boldsymbol{z}) \tag{3}$$

The neural network is typically trained by minimizing the cross entropy loss between the true label sdistribution $\boldsymbol{y}$ and the output distribution as the following:

$$Loss(\boldsymbol{z}', \boldsymbol{y}) = H(\boldsymbol{y}, \boldsymbol{z}') = -\sum_i \boldsymbol{y}_i \log \boldsymbol{z}'_i \qquad i = 1, 2, \ldots, m \tag{4}$$

where $m$ is the number of the labels. In the following, we will use $y$ to denote the true label category, $\boldsymbol{y}$ to denote the one-hot distribution of $y$, $\boldsymbol{x}'$ to denote softmax($\boldsymbol{x}$), and $H(\boldsymbol{p}, \boldsymbol{q})$ to denote the cross entropy between $\boldsymbol{p}$ and $\boldsymbol{q}$, where $\boldsymbol{p}$ is the distribution that the model needs to approximate, e.g., $\boldsymbol{y}$ in (4), and $\boldsymbol{q}$ is the distribution generated by the model, e.g., $\boldsymbol{z}'$ in (4).

### 2.1 LABEL EMBEDDING NETWORK

The label embedding is supposed to represent the semantics, i.e. similarity between labels, which makes the length of each label embedding to be the number of the labels $m$. The embedding is denoted by

$$\mathbf{E} \in \mathbb{R}^{m \times m} \tag{5}$$

where $m$ is the number of the labels. Each element in a label embedding vector represents the similarity between two labels. For example, in label $y$'s embedding vector $\boldsymbol{e} = \mathbf{E}_y$, the *i-th* value represents

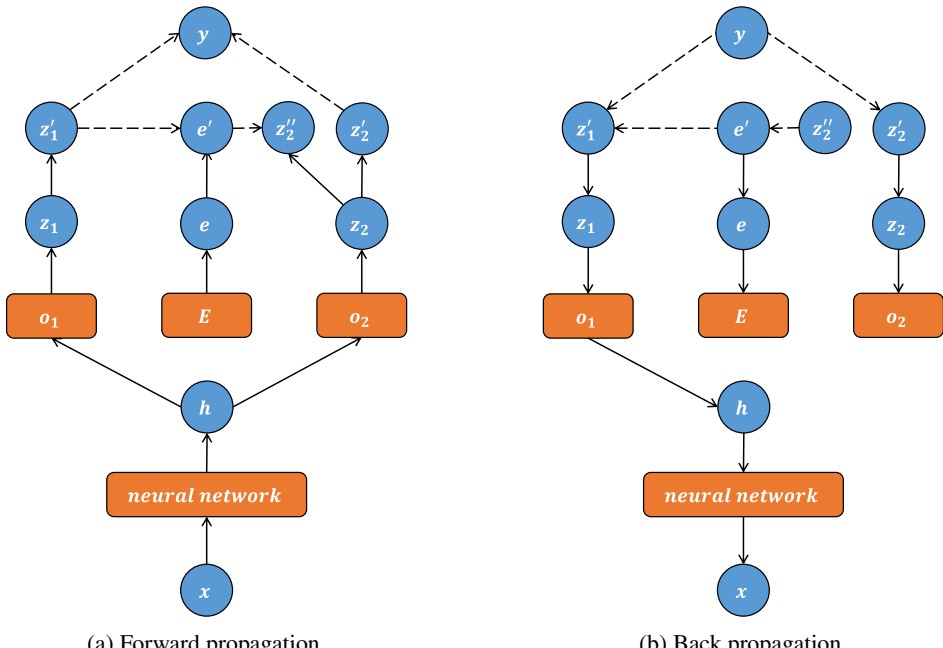

(a) Forward propagation.  (b) Back propagation.

Figure 1: Illustration of the proposed method. A circle stands for a vector, and a square stands for a layer with parameters. The dashed line means a cross entropy operation. The square labeled "neural network" can be CNN, ResNet, Seq-to-Seq, and so on.

the similarity of label $y$ to label $i$. To learn the embeddings, a reasonable approach would be to make the label embedding $e = \mathbf{E}_y$ close to the output $z$ in (4) of the neural network, whose predicted label is $y$, as the output distribution of the model contains generalization information learned by the neural network. In turn, the label embedding can be used as a more refined supervisory signal for the learning of the model.

However, the aforementioned approach affects the learning of the model, in terms of the discriminative power. In essence, the model is supposed to distinguish the inputs, while the label embedding is supposed to capture the commonness of the labels based on the inputs, and the two goals are in conflict. To avoid the conflict, we propose to separate the output representation. One output layer, denoted by $o_1$, is used to differentiate the hidden representation as normal, which is used for predicting, and the other output layer, denoted by $o_2$, focuses more on learning the similarity of the hidden representation, from which the label embedding is learned:

$$z_1 = o_1(\boldsymbol{h}), \ z_2 = o_2(\boldsymbol{h}) \tag{6}$$

The two output layers share the same hidden representation, but have independent parameters. They both learn from the one-hot distribution of the true label:

$$Loss(\boldsymbol{z}_1', \boldsymbol{y}) = H(\boldsymbol{y}, \boldsymbol{z}_1') = -\sum_i \boldsymbol{y}_i \log(\boldsymbol{z}_1')_i \qquad i = 1, 2, \ldots, m \tag{7}$$

$$Loss(\boldsymbol{z}_2', \boldsymbol{y}) = H(\boldsymbol{y}, \boldsymbol{z}_2') = -\sum_i \boldsymbol{y}_i \log(\boldsymbol{z}_2')_i \qquad i = 1, 2, \ldots, m \tag{8}$$

In back propagation, the gradient from $\boldsymbol{z}_2$ is kept from propagating to $\boldsymbol{h}$, so the learning of the $o_2$ does not affect the hidden representation. By doing this, the discriminative power of $o_1$ is maintained and even enhanced by the using of label embedding. In the meanwhile, the label embedding obtains a more stable learning target.

The label embedding is then learned by minimizing the cross entropy loss between the normalized embedding $\boldsymbol{e}' = \text{softmax}(\boldsymbol{e})$ and the normalized output $\boldsymbol{z}_2' = \text{softmax}(\boldsymbol{z}_2)$:

$$Loss(\boldsymbol{e}', \boldsymbol{z}_2') = H(\boldsymbol{z}_2', \boldsymbol{e}) = -\sum_i (\boldsymbol{z}_2')_i \log \boldsymbol{e}_i' \qquad i = 1, 2, \ldots, m \tag{9}$$

However, the above approach does not scale properly during the training, as the output $\boldsymbol{z_2}'$ becomes too close to the one-hot distribution $\boldsymbol{y}$, and the label embedding fails to capture the similarity between labels. To solve this, we apply the softmax with temperature $\tau$ to soften the distribution of the normalized $\boldsymbol{z_2}$, which is computed by

$$(\boldsymbol{z}_2'')_i = \frac{\exp((\boldsymbol{z}_2)_i/\tau)}{\sum_{j=1}^{m} \exp((\boldsymbol{z}_2)_j/\tau)} \qquad i = 1, 2, \ldots, m \tag{10}$$

and the loss becomes

$$Loss(\boldsymbol{e}', \boldsymbol{z}_2'') = H(\boldsymbol{z}_2'', \boldsymbol{e}') = -\sum_i (\boldsymbol{z}_2'')_i \log \boldsymbol{e}_i' \qquad i = 1, 2, \ldots, m \tag{11}$$

In the following we will use $\boldsymbol{z}_2''$ to denote the softmax with temperature. By applying a higher temperature, the label embedding gains more details of the output distribution, and the elements in an embedding vector other than the label-based one, i.e. the elements off the diagonal, are better learned. However, the annealed distribution also makes the difference between the incorrect labels closer. To solve the problem, we further propose to regularize the normalized output, so that the highest value of the distribution does not get too high, and the difference between labels is kept:

$$Loss(\boldsymbol{z}_2') = || \max(0, (\boldsymbol{z}_2')_y - \alpha)||_p \tag{12}$$

If $p$ equals to 1 or 2, the loss is a hinge L1 or L2 regularization. The learned embedding is in turn used in the training of the network by making the output close to the learned embedding. This is done by minimizing the cross entropy loss between the normalized output and the normalized label embedding:

$$Loss(\boldsymbol{z}_1', \boldsymbol{e}') = H(\boldsymbol{e}', \boldsymbol{z}_1') = -\sum_i \boldsymbol{e}_i' \log(\boldsymbol{z}_1')_i \qquad i = 1, 2, \ldots, m \tag{13}$$

As a fine-grained distribution of the true label is learned by the model, a faster convergence is achieved, and risk of overfitting is also reduced.

In summary, the final objective of the proposed method is as follows:

$$Loss(\boldsymbol{x}, \boldsymbol{y}; \boldsymbol{\theta}) = H(\boldsymbol{y}, \boldsymbol{z}_1') + H(\boldsymbol{e}', \boldsymbol{z}_1') + H(\boldsymbol{y}, \boldsymbol{z}_2') + || \max(0, (\boldsymbol{z}_2')_y - \alpha)||_p + H(\boldsymbol{z}_2'', \boldsymbol{e}') \tag{14}$$

Figure 1 shows the overall architecture of the proposed method. Various kinds of neural networks are compatible to generate the hidden representation. In our experiments, we used CNN, ResNet, and Seq-to-Seq. However, the choice may not be limited to those architectures. Moreover, although the output architecture is significantly re-designed, the computational cost does not increase much, as the added operations are relatively cheap in computation.

## 2.2 Compressed Label Embedding Network

When there is a massive number of labels (e.g., over 20,000 labels for neural machine translation), the embedding $\mathbf{E}$ takes too much memory. Suppose we have a neural machine translation task with 50,000 labels, then the label embedding is a $50{,}000 \times 50{,}000$ matrix. The embedding matrix alone will take up approximately 9,536MB, which is not suitable for GPU. To alleviate this issue, we propose to re-parameterize the embedding matrix to a product of two smaller matrices, $\mathbf{A}$ and $\mathbf{B}$:

$$\mathbf{A} \in \mathbb{R}^{m \times h}, \ \mathbf{B} \in \mathbb{R}^{h \times m} \tag{15}$$

where $m$ is the number of the labels, and $h$ is the size of the "compressed" label embedding. The label embedding for label $y$ is computed as the following:

$$\boldsymbol{e} = \text{ReLU}(\mathbf{A}_y \mathbf{B}) \tag{16}$$

where $\mathbf{A}_y$ means taking out the $y$-th row from the matrix $\mathbf{A}$. The resulting vector $\boldsymbol{e}$ is an $m$-dimensional vector, and can be used as label embedding to substitute the corresponding part in the final loss of a normal label embedding network. The matrix $\mathbf{A}$ can be seen as the "compressed" label embeddings, where each row represents a compressed label embedding, and the matrix $\mathbf{B}$ can be seen as the projection that reconstructs the label embeddings from the "compressed" forms. This technique can reduce the space needed to store the label embeddings by a factor of $\frac{m}{2h}$. Considering the previous example, if $h = 100$, the space needed is reduced by 250x, from 9,536MB to about 38.15MB.

Table 1: Statistics of the tasks.

| DataSet | Training | Dev | Test | Labels | Model |
|---|---|---|---|---|---|
| CIFAR-100 | 50,000 | – | 10,000 | 100 | ResNet-18 |
| CIFAR-10 | 50,000 | – | 10,000 | 10 | ResNet-8 |
| MNIST | 55,000 | 5,000 | 10,000 | 10 | CNN |
| LCSTS | 2,400,591 | 10,666 | 1,106 | 4,000 | Seq-to-Seq |
| IWSLT2015 | 133,317 | 1,268 | 1,553 | 22,439 | Seq-to-Seq |

## 3 EXPERIMENTS

We conduct experiments using different models (CNN, ResNet, and Seq-to-Seq) on diverse tasks (computer vision tasks and natural language processing tasks) to show that the proposed method is general-purpose and works for different types of deep learning models.

**CIFAR-100:** The CIFAR-100 (Krizhevsky & Hinton, 2009) dataset consists of 60,000 32×32 color images in 100 classes containing 600 images each. The dataset is split into 50,000 training images and 10,000 test images. Each image comes with a "fine" label (the class to which it belongs) and a "coarse" label (the superclass to which it belongs).

**CIFAR-10:** The CIFAR-10 dataset (Krizhevsky & Hinton, 2009) has the same data size as CIFAR-100, that is, 60,000 32×32 color images, split into 50,000 training images and 10,000 test images, except that it has 10 classes with 6,000 images per class.

**MNIST:** The MNIST handwritten digit dataset (LeCun et al., 1998) consists of 60,000 28×28 pixel gray-scale training images and additional 10,000 test examples. Each image contains a single numerical digit (0-9). We select the first 5,000 images of the training images as the development set and the rest as the training set.

**Social Media Text Summarization Dataset (LCSTS):** LCSTS Hu et al. (2015) consists of more than 2,400,000 social media text-summary pairs. It is split into 2,400,591 pairs for training, 10,666 pairs for development data, and 1,106 pairs for testing. Following (Hu et al., 2015), the evaluation metric is ROUGE-1, ROUGE-2 and ROUGE-L (Lin & Hovy, 2003).

**IWSLT 2015 English-Vietnam Dataset (IWSLT2015):** The dataset is from the International Workshop on Spoken Language Translation 2015. The dataset consists of about 136,000 English-Vietnam parallel sentences, constructed from the TED captions. It is split into training set, development set and test set, with 133,317, 1,268 and 1,553 sentence pairs respectively. The evaluation metric is BLEU score (Papineni et al., 2002).

### 3.1 EXPERIMENTAL SETTINGS

For CIFAR-100 and CIFAR-10, we test our method based on ResNet with 18 layers and 8 layers, respectively, following the settings in He et al. (2016). For MNIST, the CNN model consists of two convolutional layers, one fully-connected layer, and another fully-connected layer as the output layer. The filter size is $5 \times 5$ in the convolutional layers. The first convolutional layer contains 32 filters, and the second contains 64 filters. Each convolutional layer is followed by a max-pooling layer. Following common practice, we use ReLU (Hahnloser et al., 2000) as the activation function of the hidden layers.

For LCSTS and IWSLT2015, we test our approach based on the sequence-to-sequence model. Both the encoder and decoder are based on the LSTM unit, with one layer for LCSTS and two layer for IWSLT2015. Each character or word is represented by a random initialized embedding. For LCSTS, the embedding size is 400, and the hidden state size of the LSTM unit is 500. For IWSLT2015, the embedding size is 512, and the hidden state size of the LSTM unit is 512. We use beam search for IWSLT2015, and the beam size is 10. Due to the very large label sets, we use the compressed label embedding network (see Section 2.2) for both tasks.

Although there are several hyper-parameters introduced in the proposed method, we use a very simple setting for all tasks, because the proposed method is robust in our experiments, and simply works well without fine-tuning. We use temperature $\tau = 2$ for all the tasks. For simplicity, we use the L1 form of the hinge loss of $o_2$, and $\alpha$ is set to 0.9 for all the tasks. We use the Adam optimizer

Table 2: Results of Label Embedding on computer vision.

| CIFAR-100 | Test Error (%) | Error Reduction | Time/Epoch (s) |
|---|---|---|---|
| ResNet-18 | 27.35 (±0.40) | -3.38 (↓ **12.4**%) | 86.3 |
| ResNet-LabelEmb-18 | **23.97** (±0.33) | | 87.8 |

| CIFAR-10 | Test Error (%) | Error Reduction | Time/Epoch (s) |
|---|---|---|---|
| ResNet-8 | 8.66 (±0.43) | -1.69 (↓ **19.5**%) | 35.1 |
| ResNet-LabelEmb-8 | **6.97** (±0.22) | | 42.5 |

| MNIST | Test Error (%) | Error Reduction | Time/Epoch (s) |
|---|---|---|---|
| CNN | 0.85 (±0.11) | -0.30 (↓ **35.3**%) | 4.60 |
| CNN-LabelEmb | **0.55** (±0.03) | | 5.46 |

(a) CIFAR-100 on ResNet-18.    (b) CIFAR-10 on ResNet-8.    (c) MNIST on CNN.

Figure 2: Error rate curve for CIFAR-100, CIFAR-10, and MNSIT. 20 times experiments (the light color curves) are conducted for credible results both on the baseline and our proposed model. The average results are shown as deep color curves.

(Kingma & Ba, 2014) for all the tasks, using the default hyper-parameters. For CIFAR-100, we divide the learning rate by 5 at epoch 40 and epoch 80. As shown in the previous work (He et al., 2016), dividing the learning rate at certain iterations proves to be beneficial for SGD. We find that the technique also applies to Adam. We do not apply this technique for CIFAR-10 and MNIST, because the results are similar with or without the technique. The experiments are conducted using *INTEL Xeon 3.0GHz CPU* and *NVIDIA GTX 1080 GPU*. We run each configuration 20 times with different random seeds for the CV tasks. For the tasks without development sets, we report the results at the final epoch. For the ones with development sets, we report the test results at the epoch that achieves the best score on development set.

## 3.2 RESULTS ON COMPUTER VISION

First, we show results on CIFAR-100 and CIFAR-10, which are summarized in Table 2. As we can see, the proposed method achieves much better results. On CIFAR-100, the proposed method achieves 12.4% error reduction ratio from the baseline (ResNet-18). On CIFAR-10, the proposed method achieves 19.5% error reduction ratio from the baseline (ResNet-8). The training time per epoch is similar to the baselines. The results of MNIST are summarized in Table 2. As we can see, the proposed method achieves the error rate reduction of over 32%.

The detailed error rate curves are shown in Figure 2. The 20 repeated runs are shown in lighter color, and the averaged values are shown in deeper color. As we can see from Figure 2, the proposed method achieves better convergence speed than the ResNet and CNN baselines. This is because the label embedding achieves soft training of the model, where the conflict of the features of similar labels are alleviated by the learned label embeddings. The learned label embeddings enables the model to share common features when classifying the similar labels, because the supervisory signal contains the information about similarity, thus making the learning easier. Besides, the model is not required to distinguish the labels completely, which avoids unnecessary subtle update of the parameters.

In addition, we can see that by using label embedding the proposed method has much more stable training curves than the baselines. The fluctuation of the proposed method is much smaller than the baselines. As the one-hot distribution forces the label to be completely different from others, the original objective seeks unique indicators for the labels, which are hard to find and prone to

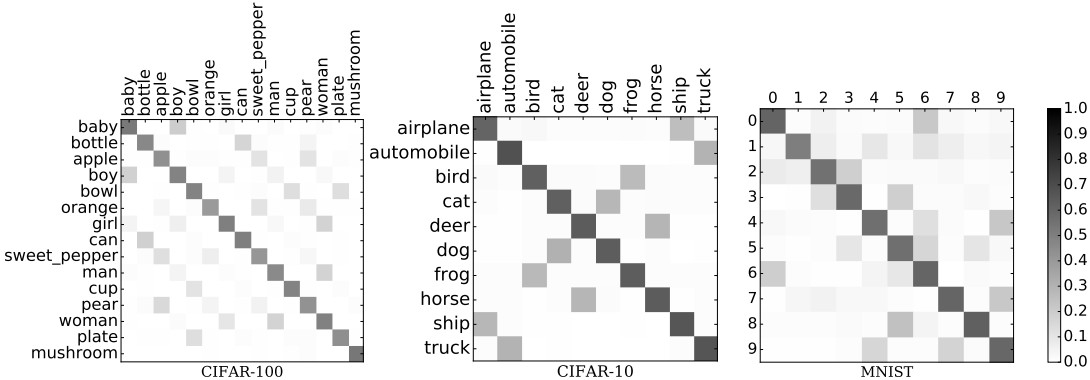

Figure 3: Heatmaps generated by the label embeddings.

Table 3: Results of Label Embedding for LCSTS (W: Word model; C: Character model). The evaluation metric is ROUGE score (higher is better).

| LCSTS | ROUGE-1 | ROUGE-2 | ROUGE-L |
|---|---|---|---|
| Seq2seq (W) (Hu et al., 2015) | 17.7 | 8.5 | 15.8 |
| Seq2seq (C) (Hu et al., 2015) | 21.5 | 8.9 | 18.6 |
| Seq2seq-Attention (W) (Hu et al., 2015) | 26.8 | 16.1 | 24.1 |
| Seq2seq-Attention (C) (Hu et al., 2015) | 29.9 | 17.4 | 27.2 |
| Seq2seq-Attention (C) (our implementation) | 30.1 | 17.9 | 27.2 |
| **Seq2seq-Attention-LabelEmb (C) (our proposal)** | **31.7 (+1.6)** | **19.1 (+1.2)** | **29.1 (+1.9)** |

overfitting, thus often leading the training astray. The proposed method avoids that by softening the target distribution, so that the features used are not required to be unique, and more common but essential features can be selected, which stabilizes the learning compared to the original objective.

The proposed method achieves comparable or even better results than the state-of-the-art systems. More detailed comparisons to the high performance systems are in Appendix D.

### 3.2.1 LEARNED LABEL EMBEDDINGS

It would be interesting to check the learned label embeddings from those datasets. Figure 3 shows the learned label embeddings from the CIFAR-100, CIFAR-10, and MNIST tasks, respectively.

For the CIFAR-100 task, as we can see, the learned label embeddings are very interesting. Since we don't have enough space to show the heatmap of all of the 100 labels, we randomly selected three groups of labels, with 15 labels in total. For example, the most similar label for the label "bottle" is "can". For the label "bowl", the two most similar labels are "cup" and "plate". For the label "man", the most similar label is "woman", and the second most similar one is "boy".

For the CIFAR-10 task, as we can see, the learned label embeddings are also meaningful. For example, the most similar label for the label "automobile" is "truck". For the label "cat", the most similar label is "dog". For the label "deer", the most similar label is "horse". For the MINST task, there are also interesting patterns on the learned label embeddings. Those heatmaps of the learned labels demonstrate that our label embedding learning is reasonable and can indeed reveal rational similarities among diversified labels. The learned embedding can also be used to directly trained a new model on the same task, with improved accuracy and faster convergence, which we will show in Appendix C.

### 3.3 RESULTS ON NATURAL LANGUAGE PROCESSING

First, we show experimental results on the LCSTS text summarization task. The results are summarized in Table 3. The performance is measured by ROUGE-1, ROUGE-2, and ROUGE-L. As we can see, the proposed method performs much better compared to the baselines, with ROUGE-1 score of 31.7, ROUGE-2 score of 19.1, and ROUGE-L score of 29.1, improving by 1.6, 1.2, and 1.9, respectively. In addition, the results of the baseline implemented by ourselves are competitive

Table 4: Results of Label Embedding for IWSLT2015. The evaluation metric is BLEU score (higher is better).

| IWSLT2015 | BLEU |
|---|---|
| Stanford NMT (Luong & Manning, 2015) | 23.3 |
| NMT (greedy) (Luong et al., 2017) | 25.5 |
| NMT (beam=10) (Luong et al., 2017) | 26.1 |
| Seq2seq-Attention (beam=10) | 25.7 |
| **Seq2seq-Attention-LabelEmb (beam=10)** | **26.8** (**+1.1**) |

Table 5: Examples of the similarity results on IWSLT2015, based on the learned label embeddings.

| Label (Word) | Top 5 Most Similar Labels |
|---|---|
| chó (dog) | cún (dogs), mèo (cat), con (baby), chú (uncle), heo (pig) |
| trai (boy) | gái (girl) , bé (little), người (people), con (children), cậu (you) |
| chạy (run) | hoạt (activity) , vận (campaign) , đi (go) , điều (thing) , làm (do) |
| hát (sing) | nhạc (music) , diễn (acting), nói (say) , học (learn) , viết (write) |
| đẹp (beautiful) | vẽ (draw) , xinh (pretty) , tuyệt (great) , hơn (than) , xắn (lovely) |
| tốt (good) | giỏi (great) , hay (or) , tuyệt (Great) , rất (very) , có (have) |
| đùa (joke) | trò (game) , cười (laugh) , chuyện (matter), chơi (play) , nói (say) |
| đỏ (red) | màu (color) , red (red) , xanh (blue) , đen (black) , vàng (yellow) |
| biển (sea) | đáy (bottom), nước (water), đại (ocean), khơi(sea) , dưới (bottom) |
| mưa (rain) | bão (storm) , trời (sky) , gió (wind) , cơn (storm) , nước (water) |

with previous work (Hu et al., 2015). In fact, in terms of all of the three metrics, our implementation consistently beats the previous work, and the proposed method could further improve the results.

Then, we show experimental results on the IWSLT2015 machine translation task. The results are summarized in Table 4. We measure the quality of the translation by BLEU, following common practice. The proposed method achieves better BLEU score than the baseline, with an improvement of 1.1 points. To our knowledge, 26.8 is the highest BLEU achieved on the task, surpassing the previous best result 26.1 (Luong et al., 2017). From the experimental results, it is clear that the compressed label embedding can improve the results of the Seq-to-Seq model as well, and works for the tasks, where there is a massive number of labels.

### 3.3.1 LEARNED LABEL EMBEDDINGS

The label embedding learned in compressed fashion also carries semantic similarities. We report the sampled similarities results in Table 5. As shown in Table 5, the learned label embeddings capture the semantics of the label reasonably well. For example, the word "đỏ" (red) is most similar to the colors, i.e. "màu" (color), "red" (red), "xanh" (blue), "đen" (black), and "vàng" (yellow). The word "mưa (rain)" is most similar to "bão" (storm), "trời" (sky), "gió" (wind), "cơn" (storm), "nước" (water), which are all semantically related to the natural phenomenon "rain". The results of the label embeddings learned in a compressed fashion demonstrate that the re-parameterization technique is effective in saving the space without degrading the quality of the learned label embeddings. They also prove that the proposed label embedding also works for NLP tasks.

## 4 CONCLUSION

We propose a method that can learn label representation during the training process of deep neural networks. Furthermore, we propose a solution to automatically learn compressed label embedding, such that the memory cost is substantially reduced. The proposed method can be widely applied to different models. We conducted experiments on CV tasks including CIFAR-100, CIFAR-10, and MNIST, and also on natural language processing tasks including LCSTS and IWSLT2015. Results suggest that the proposed method achieves significant better accuracies than the existing methods (CNN, ResNet, and Seq-to-Seq). Moreover, the learned label embeddings are reasonable and interpretable, which provides meaningful semantics of the labels. We achieve comparable or even better results with the state-of-the-art systems on those tasks.

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

## A    ADDITIONAL CONSIDERATIONS OF THE PROPOSED METHOD

To achieve good performance, there are some additional considerations for the proposed method.

First, when learning the label embedding, if the current output from the model is wrong, which often happens when the training just begins, the true label's embedding should not learn from the output from the model. This is because the information is incorrect to the learning of the label embedding, and should be neglected. This consideration can be particularly useful to improve the performance under the circumstances where the model's prediction is often wrong during the start of the training, e.g. the CIFAR-100 task and the neural machine translation task.

Second, we suggest using the diagonal matrix as the initialization of the label embedding matrix. By using the diagonal matrix, we provide a prior to the label embedding that one label's embedding should be the most similar to the label itself, which could be useful at the start of the training and beneficial for the learning.

## B    EXPERIMENTS OF MLP ON MNIST

Table 6: Results of Label Embedding for MNIST using MLP.

| **MNIST** | Dev Error (%) | Test Error (%) | Test Error Reduction | Time/Epoch (s) |
|---|---|---|---|---|
| MLP | 1.81 | 1.93 (±0.27) | **-0.50 (↓ 25.9%)** | 1.65 |
| MLP-LabelEmb | **1.29** | **1.43** (±0.06) | | 2.84 |

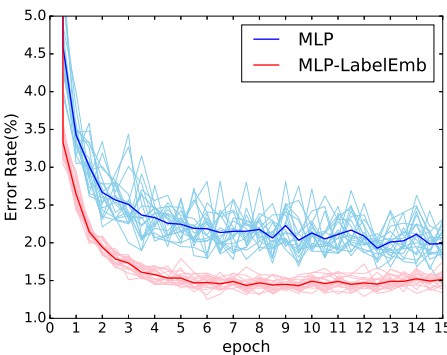

Figure 4: Error rate curve for MLP model on MNSIT. 20 times experiments (the light color curves) are conducted for credible results both on the baseline and our proposed model. The average results are shown as deep color curves.

We also conducted experiments on MNIST, using the MLP model. The MLP model consists of two 500-dimensional hidden layers and one output layer. The other settings are the same as the CNN model.

The experimental results are summarized in Table 6. As we can see, the proposed label embedding method achieves better performance than the baseline, with an error rate reduction over 24%. All the results are the averaged error rates over 20 repeated experiments, and the standard deviation results are also shown.

Figure 4 shows the detailed error rate curve of the MLP model. The 20 repeated runs are shown in light color, and the averaged values are shown in deeper color. As shown, the proposed method also works for MLP, and the results are consistently better than the baselines. As the same with the CNN model, the proposed method converges faster than the baseline.

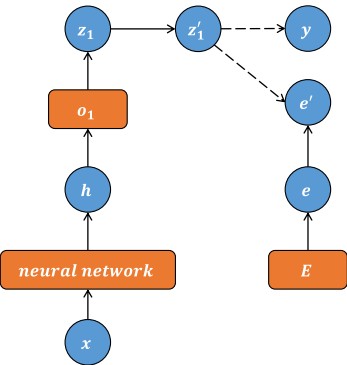

Figure 5: Illustration of the forward propagation of Pre-Trained Label Embedding Network.

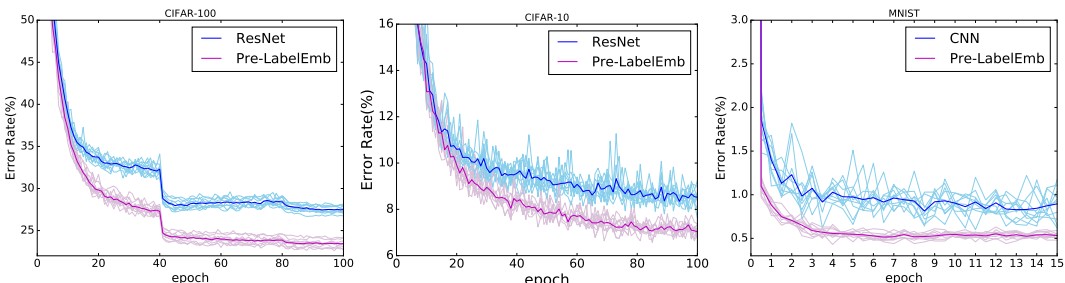

Figure 6: Error rate curve for CIFAR-100, CIFAR-10 and MNIST. Pre-trained label embeddings are used.

## C   LEARNED LABEL EMBEDDING IS USEFUL

In the following section, we will show that the learned label embedding is not only reasonable, but also useful for applications. For example, the learned label embedding can be directly used as fine-grained true label distribution to train a new model on the same dataset. For this purpose, the new model's objective function contains two parts, i.e., the original one-hot label based cross entropy objective, together with a label embedding based cross entropy objective. We call this model Pre-Trained Label Embedding Network. The Label Embedding Network means that the network uses label embedding to improve the training of the network, and the difference of the pre-trained label embedding network from the one presented in Section 2.1 is that in the pre-trained label embedding network, the label embedding is pre-trained and fixed, thus eliminating the need for learning the embedding, while in the label embedding network, the label embedding is learned during the training. In implementation, there are two main differences. First, the label embedding $\mathbf{E}$ is fixed and requires no learning. Second, the sub-network $o_2$, which learns the label embedding, is removed — because there is no need to learn the label embedding again. Thus, the pre-trained label embedding network has the loss function as follows:

$$Loss(\boldsymbol{x}, \boldsymbol{y}; \boldsymbol{\theta}) = H(\boldsymbol{y}, \boldsymbol{z}_1') + H(\boldsymbol{e}', \boldsymbol{z}_1') \tag{17}$$

The pre-trained label embedding network is illustrated in Figure 5.

Figure 6 shows the results of the pre-trained label embedding network, whose label embedding is learned by a normal label embedding network. As we can see, pre-trained label embedding network can achieve much better result than the baseline, with faster convergence. It shows that the learned label embedding is effective in improving the performance of the same model and the label embedding indeed contains generalization information, which provides a more refined supervised signal to the model. In this way, the learned label embeddings can be saved and be reused to improve the training of different models on the task, and there is no need to learn the label embedding again.

# D  COMPARING WITH HIGH PERFORMANCE SYSTEMS ON THE COMPUTER VISION TASKS

For the CIFAR-100 task, the error rate is typically from 38% to 25% (Goodfellow et al., 2013; Lin et al., 2014; Springenberg et al., 2015; Srivastava et al., 2015; He et al., 2016; Romero et al., 2015; Clevert et al., 2016; Graham, 2015; Mishkin & Matas, 2016). Goodfellow et al. (2013) achieves 38.50% error rate with Maxout Network. Springenberg et al. (2015) achieves 33.71% error rate by replacing max-pooling by a convolutional layer (All-CNN-C). Srivastava et al. (2015) achieves 32.24% error rate with Highway Network. He et al. (2016) achieves 25.16% rate with a 110 layer ResNet. Our 18-layer ResNet system achieves the averaged error rate of 23.97% over 20 repeated runs. If considering a good run, our method achieves 23.25% error rate with only 18 layers.

For CIFAR-10 task, the error rate is typically from 15% to 7% (Wan et al., 2013; Lin et al., 2014; Springenberg et al., 2015; Romero et al., 2015; Liang & Hu, 2015; Srivastava et al., 2015; He et al., 2016; Clevert et al., 2016; Goodfellow et al., 2013). Further improvement can be achieved by fine-tuning the model and the optimization method (Zagoruyko & Komodakis, 2016). To show the robustness of the proposed method, we do not fine tune the hyper-parameters. We simply use a plain ResNet-8 with an Adam optimizer with default parameters. Wan et al. (2013) achieves 11.10% error rate by applying the DropConnect technique. Goodfellow et al. (2013) achieves 9.38% error rate with Maxout Network. He et al. (2016) achieves 8.75% error rate with a 20 layer ResNet, and further achieves 6.43% error rate with a 110 layer ResNet. Our 8-layer ResNet model achieves the averaged error rate 6.97% over 20 repeated runs. If considering a good run, our method achieves 6.32%.

For MNIST task, plain convolutional networks typically achieve error rates ranging widely from more than 1.1% to around 0.4% (Goodfellow et al., 2013; Simard et al., 2003; Srivastava et al., 2015). Data augmentation and other more complicated models can further improve the performance of the models (Wan et al., 2013; Ciresan et al., 2012; Graham, 2015), which we believe also work for our method. Srivastava et al. (2015) achieves 0.57% error rate by using Highway Network. Mishkin & Matas (2016) achieves 0.48% error rate by using LSUV initialization for FitNets. Our CNN model achieves the averaged error rate of 0.55%. If considering a good run, our model achieves 0.42%.

# E  DETAILED COMPARISONS TO RELATED WORK

The prior studies on label representation in deep learning are limited. Existing label representation methods are mostly on traditional methods out of deep learning frameworks. Those label representation methods also adopt the name of label embedding. However, the meaning is different from that in the sense of deep learning. Those label representation methods intend to obtain a representation function for labels. The label representation vector can be data independent or learned from existing information, including training data (Weston et al., 2011), auxiliary annotations (Akata et al., 2013), class hierarchies (Bengio et al., 2010), or textual descriptions (Ma et al., 2016). For example, in Hsu et al. (2009), the label embedding is fixed and is set independently from the data by random projections, and several regressors are used to learn to predict each of the elements of the true label's embedding, which is then reconstructed to the regular one-hot label representation for classification. Another example is the Canonical Correlation Analysis (CCA), which seeks vector $a$ and vector $b$ for random variables $X$ and $Y$, such that the correlation of the variables $a'X$ and $b'Y$ is maximized, and then $b'Y$ can be regarded as label embeddings (Hardoon et al., 2004).

There are several major differences between those methods and our proposed method. First, most of those methods are not easy to adapt to deep learning architectures. As previously introduced, those methods come with a totally different architecture and their own learning methods, which are not easy to extend to general-purpose models like neural networks. Instead, in the proposed method, label embedding is automatically learned from the data by back propagation. Second, the label representation in those methods is not adapting during the training. In Hsu et al. (2009), the label embedding is fixed and randomly initialized, thus revealing none of the semantics between the labels. The CCA method is also not adaptively learned from the training data. In all, their learned label representation lacks interaction with other model parameters, while label embeddings obtained from our proposed method both reveal the semantics of the labels and interact actively with the other parts of the model by back propagation.

There have also been prior studies on so-called "soft labels". The soft label methods are typically for binary classification (Nguyen et al., 2014), where the human annotators not only assign a label for an example, but also give information on how confident they are regarding the annotation. The side information can be used in the learning procedure to alleviate the noise from the data and produce better results. The main difference from our method is that the soft label methods require additional annotation information (e.g., the confidence information of the annotated labels) of the training data, while our method does not need additional annotation information, and the "soft" probability is learned during the training in a simple but effective manner. Moreover, the proposed method is not restricted to binary classification.

There have also been prior studies on model distillation in deep learning that uses label representation to better compress a big model into a smaller one. In deep learning, it's common sense that due to the non-convex property of the neural network functions, different initialization, different data order, and different optimization methods would cause varied results of the same model. Model distillation (Hinton et al., 2014) is a novel method to combine the different instances of the same model into a single one. In the training of the single model, its target distribution is a combination of the output distributions of the previously trained models. Our method is substantially different compared with the model distillation method. The motivations and designed architectures are both very different. The model distillation method adopts a pipeline system, which needs to first train a large model or many different instances of models, and then use the label representation of the baseline models to provide better supervisory signals to re-train a smaller model. This pipeline setting is very different from our single-pass process setting. Our method also enables the ability to learn compressed label embedding for an extremely large number of labels. Moreover, for a given label, the label representation in their method is different from one example to another. That is, they do not provide a universal label representation for a label, which is very different compared with our setting.

