# OpenReview forum: "Label Embedding Network: Learning Label Representation for Soft Training of Deep Networks"
_ICLR.cc/2018/Conference — Reject_

### Official Review · AnonReviewer1 · 2017-11-24

**Rating:** 4
**Confidence:** 5

**Review:**

The paper proposes to add an embedding layer for labels that constrains normal classifiers in order to find label representations that are semantically consistent. The approach is then experimented on various image and text tasks.

The description of the model is laborious and hard to follow. Figure 1 helps but is only referred to at the end of the description (at the end of section 2.1), which instead explains each step without the big picture and loses the reader with confusing notation. For instance, it only became clear at the end of the section that E was learned.

One of the motivations behing the model is to force label representations to be in a semantic space (where two labels with similar meanings would be nearby). The assumption given in the introduction is that softmax would not yield such a representation, but nowhere in the paper this assumption is verified. I believe that using cross-entropy with softmax should also push semantically similar labels to be nearby in the weight space entering the softmax. This should at least be verified and compared appropriately.

Another motivation of the paper is that targets are given as 1s or 0s while soft targets should work better. I believe this is true, but there is a lot of prior work on these, such as adding a temperature to the softmax, or using distillation, etc. None of these are discussed appropriately in the paper.

Section 2.2 describes a way to compress the label embedding representation, but it is not clear if this is actually used in the experiments. h is never discussed after section 2.2.

Experiments on known datasets are interesting, but none of the results are competitive with current state-of-the-art results (SOTA), despite what is said in Appending D. For instance, one can find SOTA results for CIFAR100 around 16% and for CIFAR10 around 3%. Similarly, one can find SOTA results for IWSLT2015 around 28 BLEU. It can be fine to not be SOTA as long as it is acknowledged and discussed appropriately.

---

### Official Review · AnonReviewer3 · 2017-11-27
**not a well presented/justified  model**

**Rating:** 4
**Confidence:** 3

**Review:**

This paper proposes a label embedding network method that learns label embeddings during the training process of deep networks.
Pros: Good empirical results.
Cons:  There is not much technical contribution. The proposed approach is neither well motivated, nor well presented/justified.  The presentation of the paper needs to be improved.

1. Part of the motivation on page 1 does not make sense. In particular, for paragraph 3, if the classification task is just to separate A from B, then (1,0) separation should be better than (0.8, 0.2).

2. Label embedding learning has been investigated in many previous works. The authors however ignored all the existing works on this topic, but enforce label embedding vectors as similarities between labels in Section 2.1 without clear motivation and justification. This assumption is not very natural — though label embeddings can capture semantic information and label correlations, it is unnecessary that label embedding matrix should be m xm and each entry should represent the similarity between a pair of labels.  The paper needs to provide a clear rationale/justification for the assumptions made, while clarifying the difference (and reason) from the literature works.

3. The proposed model is not well explained.
(1) By using the objective in eq.(14), how to learn the embeddings E?
(2) The authors state “In back propagation, the gradient from z2 is kept from propagating to h”.  This makes the learning process quite arbitrary under the objective in eq.(14).
(3) The label embeddings are not directly used for the classification (H(y, z’_1)), but rather as auxiliary part of the objective.  How to decide the test labels?

---

### Official Review · AnonReviewer2 · 2017-11-29
**Technique not properly justified; not enough insights can be learned from the work.**

**Rating:** 3
**Confidence:** 4

**Review:**

The paper proposes a method which jointly learns the label embedding (in the form of class similarity) and a classification model. While the motivation of the paper makes sense, the model is not properly justified, and I learned very little after reading the paper.

There are 5 terms in the proposed objective function. There are also several other parameters associated with them: for example, the label temperature of z_2’’ and and parameter alpha in the second last term etc.

For all the experiments, the same set of parameters are used, and it is claimed that “the method is robust in our experiment and simply works without fine tuning”. While I agree that a robust and fine-tuning-free model is ideal 1) this has to be justified by experiment. 2) showing the experiment with different parameters will help us understand the role each component plays. This is perhaps more important than improving the baseline method by a few point, especially given that the goal of this work is not to beat the state-of-the-art.

---

### Public Comment · (anonymous) · 2017-11-30
**There is a similar work**

The authors do not mention a similar recent paper:
https://arxiv.org/abs/1609.06693

---

### Decision · Program_Chairs · 2018-01-29
**ICLR 2018 Conference Acceptance Decision**

**Decision:**

Reject

**Comment:**

This paper proposes an approach for jointly learning a label embedding and prediction network, as a way of taking advantage of relationships between labels.  This general idea is well-motivated, but the specifics of the proposed approach are not motivated or described well.  More discussion of relationship with prior work (e.g. other ways of "softening" the softmax) is needed.  The authors claim to have state-of-the-art results, but reviewers point out that much better results exist.